# Systematic Comparison of Tsunami Simulations on the Chilean Coast Based on Different Numerical Approaches

Sven Harig [1,*], Natalia Zamora [2], Alejandra Gubler [3,4] and Natalja Rakowsky [1]

1   Alfred Wegener Institute, Helmholtz Centre for Polar and Marine Research, 27570 Bremerhaven, Germany; natalja.rakowsky@awi.de
2   Barcelona Supercomputing Center, 08034 Barcelona, Spain; natalia.zamora@bsc.es
3   Programa de Doctorado en Geografía, Instituto de Geografía, Pontificia Universidad Católica de Chile, Santiago 7820436, Chile; agubler@uc.cl
4   Research Center for Integrated Disaster Risk Management (CIGIDEN), Santiago 7820436, Chile
*   Correspondence: sven.harig@awi.de

**Abstract:** Tsunami inundation estimates are of crucial importance to hazard and risk assessments. In the context of tsunami forecast, numerical simulations are becoming more feasible with the growth of computational power. Uncertainties regarding source determination within the first minutes after a tsunami generation might be a major concern in the issuing of an appropriate warning on the coast. However, it is also crucial to investigate differences emerging from the chosen algorithms for the tsunami simulations due to a dependency of the outcomes on the suitable model settings. In this study, we compare the tsunami inundation in three cities in central Chile (Coquimbo, Viña del Mar, and Valparaíso) using three different models (TsunAWI, Tsunami-HySEA, COMCOT) while varying the parameters such as bottom friction. TsunAWI operates on triangular meshes with variable resolution, whereas the other two codes use nested grids for the coastal area. As initial conditions of the experiments, three seismic sources (2010 *Mw* 8.8 Maule, 2015 *Mw* 8.3 Coquimbo, and 1730 *Mw* 9.1 Valparaíso) are considered for the experiments. Inundation areas are determined with high-resolution topo-bathymetric datasets based on specific wetting and drying implementations of the numerical models. We compare each model's results and sensitivities with respect to parameters such as bottom friction and bathymetry representation in the varying mesh geometries. The outcomes show consistent estimates for the nearshore wave amplitude of the leading wave crest based on identical seismic source models within the codes. However, with respect to inundation, we show high sensitivity to Manning values where a non-linear behaviour is difficult to predict. Differences between the relative decrease in inundation areas and the Manning *n*-range (0.015–0.060) are high (11–65%), with a strong dependency on the characterization of the local topo-bathymery in the Coquimbo and Valparaíso areas. Since simulations carried out with such models are used to generate hazard estimates and warning products in an early tsunami warning context, it is crucial to investigate differences that emerge from the chosen algorithms for the tsunami simulations.

**Keywords:** tsunami modelling; numerical codes comparison; inundation uncertainties; Manning values; central Chile

## 1. Introduction

Tsunamis are rare events that can pose a major threat to coastal communities. Large tsunami events have devastated coastal regions and produced large numbers of victims, as has been recorded in the last decades [1]. As a consequence, mitigation efforts on a global and local scale have increased awareness with regard to the tsunami risk and the development of hazard maps and early warning systems, which are now operational in many highly affected regions of the world.

These systems aim at predicting the potential wave impact and arrival times based on observations such as seismic or ocean recordings, which indicate that a tsunami might have

been triggered offshore. The time available for early warning can be extremely short; at the same time, observations such as tide gauges and buoy recordings are sparse, and thus numerical models play an important role in the estimations needed to activate evacuation protocols [2–4].

In the early warning context, and in tsunami hazard assessment in general, crucial information such as arrival times and estimated wave height are frequently based on numerical modelling. Although uncertainties with respect to the tsunami source might pose the largest problem for reliable hazard forecasting, it is still important to estimate uncertainties due to the numerical set-up used to generate the warning products [5,6] and the probabilistic tsunami hazard assessment [7,8]. For instance, typical model results used in a general early warning context and disseminated to the public by national and regional warning centres are estimations of the arrival times and coastal wave amplitudes to be expected in the course of the event in selected forecast points. Since the tsunami propagation in the deep ocean is well-described by linear approximations, the necessary quantities are often approximated with the use of rather coarse models solving the linearized shallow water equations. In this case, computations may be carried out very efficiently, but estimates at the coastline need special attention since in coarse simulations particularly, the simplified equations do not allow for calculations right up to the coast, and projections from model estimates offshore need to be applied (also refer to Glimsdal et al. [9]).

Thus, tsunami numerical models have been made available to the scientific and technical communities in the last decades, aiding in tsunami hazard long-term assessment and for the short-term tsunami early warning operations. Their computational demands have been met owing to the improvements in high-performance computations [10]. Highly parallel applications and the use of platforms accelerated by a graphics processing unit (GPU) and the field-programmable gate arrays (FPGA) components considerably increase the number of calculations per unit time, enabling experts to resolve inundation in high-resolution domains. These developments allow for the fast calculation of wave propagation models, including computationally expensive processes such as inundation, thus potentially qualifying such calculations for real-time emergency response workflows. For instance, the COMCOT code [11] has been parallelized and implemented in several studies, e.g., to forecast tsunami hazard [12–14]. The Tsunami-HYSEA code has passed through several benchmarks such as the US National Tsunami Hazard Mitigation Program (NTHMP), and it has been implemented in many Tsunami Early Warning Centres. It has also contributed to fast numerical modelling based on GPU [15] for exascale simulations, allowing for faster than real-time tsunami forecasting [16]. On the other hand, TsunAWI was originally designed to pre-compute scenarios and store the warning products in a database for fast access in case of a real event (see also [6]). Consequently, OpenMP was chosen for parallelization. Recently, as part of the project "Large-Scale Execution for Industry and Society" (LEXIS, EU Horizon 2020), TsunAWI was prepared for a real-time simulation, including inundation. First, the focus was on regional simulations and optimal mesh resolution [17]; later, MPI parallelization was added such that real-time simulations were no longer restricted to regional set-ups [18]. However, despite these advances, some aspects of parametrization such as the bottom roughness are often overlooked in the tsunami hazard assessment and inundation numerical modelling workflow, as discussed by Behrens et al. [19].

The tsunami propagation in the deep ocean is sufficiently well-estimated by numerical schemes with focus on the linear balance between acceleration and the pressure gradient force. Instead, in the near-shore range the non-linear components of the governing equations gain in relevance [20,21]. Especially in the inundation area, the roughness parameters play an important role in the extent of tsunami inundation. In particular, friction greatly affects the velocities and the momentum flux [22], which is relevant in structural engineering. The quality of bathymetric and topographic data as well as bottom friction parametrization and the numerical implementation in tsunami codes is crucial for model predictions [22,23]. In this context, the study by Griffin et al. [24] investigated the relevance of topography data

in tsunami modeling. Bricker et al. [25] concluded that along urban areas, the commonly used Manning's values may be too small for tsunami numerical simulations.

In this study, we conduct a systematic comparison of three tsunami numerical codes. We investigate the outcomes of the models TsunAWI, Tsunami-HySEA, and COMCOT offshore from central Chile, with specific inundation studies in Valparaíso, Viña del Mar, and Coquimbo. Furthermore, we compare the modelling results and sensitivities of the numerical codes with respect to spatial resolution and the bottom friction as well as bathymetry representation in the varying mesh geometries. We consider models for the complete life cycle of the tsunami propagation, from the source area to the coastal inundation. In such simulations, the actual numerical implementation becomes more crucial since the shallow-water theory needs to be augmented by a run-up scheme. In this context, our scope is to determine how far the numerical implementation influences the tsunami properties, varying the relevant parameters such as bottom relief and roughness. This investigation is part of the tsunami component of the RIESGOS project, which deals more generally with multi-hazard assessments in the Andes region (see www.riesgos.de). This project contributes to the wider context of risk assessment by also considering damage and loss assessments derived from results of earthquake and tsunami models (see also Gomez-Zapata et al. [26]).

This paper is organized as follows: In Section 2, we describe the layout of the study more closely with details of the data and models that were used and the set-up of experiments. Section 3 contains the results of the experiments with respect to wave propagation and inundation from all model and parameter combinations covered in the study. Section 4 contains a discussion of the outcomes, and in Section 5, we draw the main conclusions.

## 2. Data and Methods

For a systematic comparison, we investigate the outcomes of the numerical simulations for sea surface elevation, run-up, and flow depth using three tsunami events with corresponding initial conditions and similar topo-bathymetry. The selection of three different sites with varied topography conditions and three different sources allows us to cover a range of experiments, not with the use of a probabilistic approach but rather with a case-specific one. It is known that simulations of tsunami inundation are greatly influenced by the available topography and bathymetry data as well as by the bottom roughness in the coastal area. Griffin et al. [24] highlighted the importance of high-quality topography data, especially with respect to the vertical accuracy, whereas Kaiser et al. [23] and Gayer et al. [27] investigated the dependence of inundation results on the bottom roughness distribution as it might be derived from land use maps. In the present study, we use the Manning form of bottom roughness parametrization. Here, a wide range of the Manning $n$ parameter is explored; which is kept spatially constant in each grid. The sensitivity of quantities such as the inundation area with respect to $n$ is also investigated, as well as the outcomes of three codes and their required topo-bathymetry. In summary, we compare the consistency of the results of the numerical models with respect to:

- Virtual tide gauges in various water depths as well as a number of reference inland gauges;
- Tide gauge locations and records, where available;
- Flow depth on land and run-up height.

### 2.1. Spatial Domains and Mesh Resolution

In particular, we focus on high-resolution tsunami inundation in the coastal cities of Coquimbo and Valparaíso. The General Bathymetric Chart of the Oceans [28] is used in the main domain (Grid 1, 15° S–45° S) and on the second level of the nested grids (Figure 1). Levels 3 and 4 are modified from those grids used by Zamora et al. [29], originally constructed from data provided by the Hydrographic and Oceanographic Service of the Chilean Navy (SHOA).

Bathymetry and topography values are implemented as consistently as possible, given the different mesh topologies. The nested grids and triangular mesh needed for

different numerical models may vary considerably in the deep ocean; however, we assure a comparable resolution in the coastal areas. The three experiments are historical tsunami events and their corresponding records, and field observations are only used as reference for the models since we do not aim for seismic scenarios nor code validation.

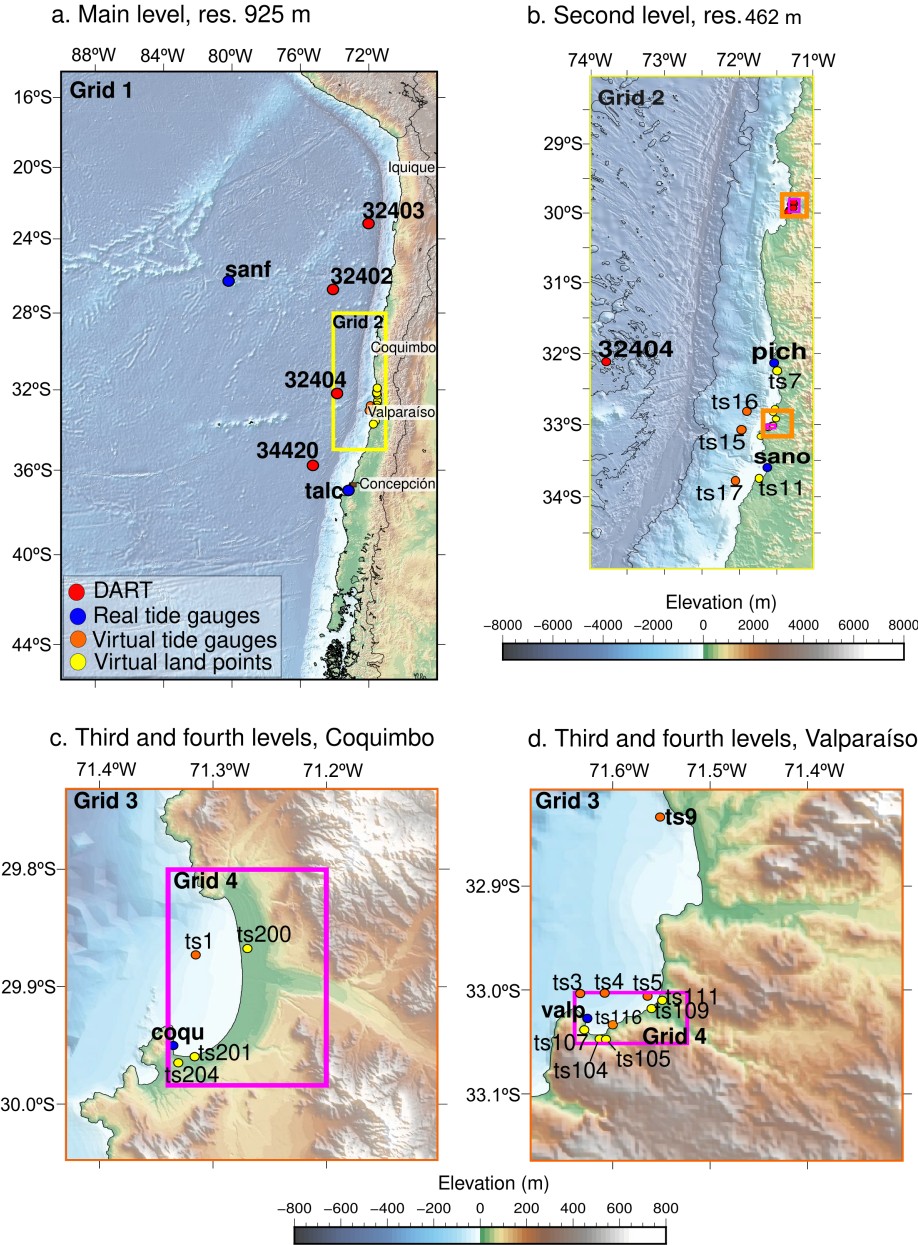

**Figure 1.** Bathymetry domains used for the simulations with Tsunami-HySEA and COMCOT. (**a**). Largest domain (grid level 1), where mostly DART and tide gauges are shown in bold. Red circles stand for Deep-ocean Assessment and Reporting of Tsunamis (DART), blue circles refer to real tide gauges, and the orange and yellow circles show virtual offshore and inland gauges, respectively. (**b**). Second level of the nested grids. (**c**). Third and fourth levels of the nested grids are shown for the Coquimbo set-up. (**d**). Third and fourth levels of the nested grids are shown for the Valparaíso set-up.

The model domain is discretized according to the format required by each numerical method. The models Tsunami-HySEA and COMCOT operate in identical sets of nested grids, the resolutions of which are listed in Table 1. TsunAWI, however, uses a triangular

mesh with an edge length prescribed by a criterion based on the CFL number that was inferred from the phase velocity and the steepness of the bathymetry slope:

$$\Delta x \approx \min\left\{ c_1 \sqrt{gh}, c_2 \frac{h}{\nabla h} \right\} \tag{1}$$

where $h$ denotes the depth of the ocean with respect to the mean sea level, and $c_1$, $c_2$ are constants specified by the coarsest resolution, which are to be chosen empirically. The mean resolution on land is a chosen constant in the potential inundation area, derived from the terrain height and the distance from the coast. In the mesh used for the current study, the coarsest resolution is about 12 km, whereas the finest resolution is about 10 m in the areas of interest—Valparaíso, Viña del Mar, and Coquimbo. The mesh structure in a small section of the mesh in the area of Valparaíso bay is shown in Figure 2. Very high resolution is restricted to pilot areas; nevertheless, the inundation scheme is applied along the entire coast, thus enabling a consistent coastal boundary condition in TsunAWI.

**Table 1.** Set-up for numerical simulations.

| Model | Tsunami-HySEA | TsunAWI | COMCOT |
|---|---|---|---|
| Spatial discretisation | 4 nested grids | Triangular mesh | 4 nested grids |
| Resolution | 925 m, 462 m, 57 m, 7.25 m | Edge length range 12 km–10 m | 925 m, 462 m 57 m, 7.25 m |
| Time stepping | Leap frog and 2nd order TVD-WAF flux-limiter scheme 0.5 s | Leap frog 0.1 s glob | Leap frog 1.0 s, automatically adjusted to satisfy the Courant condition |
| Numerical approach | Finite Volume | Finite Elements | Finite Differences |
| Inundation scheme | TVD-weighted averaged flux (WAF) flux-limiter | Extrapolation scheme | Moving boundary |

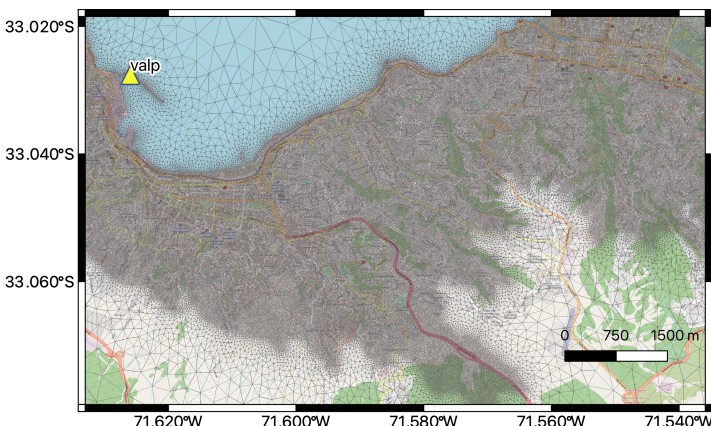

**Figure 2.** Small section of the triangular mesh used in the TsunAWI simulations. The area corresponds to Valparaíso. The yellow triangle marks the tide gauge "valp" used for model–data comparisons. Refer to Figure S1 in the Supplementary Materials, where resolution values are shown. Basemap: © OpenStreetMap Contributors.

Different numbers of computational nodes are used by the numerical models. For example, Grid 1 in Tsunami-HySEA and COMCOT is composed of 9,823,440 nodes, whereas TsunAWI uses 9,066,683 nodes in the triangular mesh. Grid 4 in Valpraíso has 1,335,552 nodes, while Coquimbo has 6,014,976 nodes.

*2.2. Numerical Models*

The models used in the comparison are based on the main numerical methods currently applied in ocean modelling: Finite differences (COMCOT), finite volumes (Tsunami-HySEA), and finite elements (TsunAWI). The governing equations solved by the models are the shallow water equations discretized in the respective numerical approach. In a general form, the equations for momentum and volume conservation are given by:

$$\partial_t \mathbf{u} + (\mathbf{u} \cdot \nabla)\mathbf{u} + \mathbf{f} \times \mathbf{u} + g\nabla\zeta - \nabla \cdot A_h \nabla\mathbf{u} + \frac{gn^2\mathbf{u}|\mathbf{u}|}{H^{4/3}} = 0 \tag{2}$$

$$\partial_t \zeta + \nabla \cdot (\mathbf{u}H) = 0 \tag{3}$$

In these equations, $\mathbf{u}$ denotes the height-averaged horizontal velocity vector, $\zeta$ stands for the sea surface elevation, $H = h + \zeta$ is the total water depth, $g = 9.81$ m/s$^2$, $\mathbf{f}$, and $A_h$ are the constant of gravity, Coriolis parameter, and horizontal viscosity coefficient, respectively. All models use bottom friction in the Manning form, exemplified in the last term of the left-hand side of Equation (2) and prescribed by the Manning parameter $n$. These Manning values were varied in each simulation to compare their effect. For simplicity, and in order to use the same capabilities of each numerical code, all domains in each simulation kept the same values. Thus, non-space-varying (constant) Manning values were tested according to each grid extent, and no spatial (cell) differences in the Manning values were considered.

The following paragraphs contain brief descriptions of the models, and the main quantities from the model approaches are summarised in Table 1.

Tsunami-HySEA (HS)

This numerical code is a part of the codes used for simulating tsunamis, developed by the EDANYA Group from the Universidad de Málaga (UMA). The code solves the two-dimensional shallow-water system using a second to third high-order finite-volume method. High order is achieved by a non-linear total variation diminishing reconstruction operator of the unknown variables such as amplitude and the vertical and horizontal velocities. At each cell interface, Tsunami-HySEA uses Godunov's method based on the approximation of 1D projected Riemann problems along the normal direction to each edge. Tsunami-HySEA also implements a two-step scheme similar to leap-frog for the deep water propagation step, and a second-order TVD-WAF flux-limiter scheme for the close-to-coast propagation-to-inundation step [15,30]. The combination of both schemes guarantees mass conservation in the complete domain and prevents the generation of spurious high-frequency oscillations near the discontinuities generated by the leap-frog type schemes (more details at https://edanya.uma.es/hysea/index.php/models/tsunami-hysea, accessed on 6 June 2022). The input topo-bathymetry needed is based on the nested grids.

COMCOT (CC)

The Cornell Multi-Grid Coupled Tsunami model (COMCOT) solves the conservative form of shallow water equations (linear and non-linear) to simulate the propagation and run-up processes of a tsunami. It is implemented with Spherical and Cartesian Coordinates. As a numerical method, COMCOT uses the explicit staggered leap-frog finite difference schemes to solve the aforementioned equations. The non-linear equation includes bottom friction effects evaluated via Manning's formula. A two-way nested grid algorithm allows to dynamically couple up to 12 levels with different grid resolutions. Grid layers should be rectangular regions with a uniform grid size. The current version used here often requires a

large stack size on the grid dimensions of a simulation, which may not be big enough based on the system's default settings (X. Wang, pers. Comm.). More details can be found in Wang and Power [11]. Similar to HS, the topo-bathymetry input needed is based on the nested grids.

TsunAWI (TSW)

This numerical code was developed in the framework of the German Indonesian Tsunami Early Warning System (GITEWS), which started in 2005 and is available for free download (https://gitlab.awi.de/tsunawi/tsunawi, accessed on 6 June 2022). Numerically, the model is based on a finite element discretization in triangular meshes. Thus, topo-bathymetry uses a bilinear interpolation of the nested grid information. The governing equations are the non-linear shallow water equations augmented by the parametrisations of bottom friction (Manning) and viscosity (Smagorinsky). The run-up scheme used in this study is based on an extrapolation method similar to the one described by Lynett et al. [31]. More information on the model is contained in the article by Rakowsky et al. [32] and on the project website (https://tsunami.awi.de, accessed on 6 June 2022).

*2.3. The Experiments: Seismic Sources*

We used three different seismic sources along the Chilean tectonic margin, extending about 1000 km from Coquimbo to Concepción. The seismic sources (finite fault models) for the 2015 *Mw* 8.3 Coquimbo earthquake, the 2010 *Mw* 8.8 Maule earthquake, and the large *Mw* 9.1 Valparaíso earthquake were used to obtain the initial sea surface elevations (Figure 3); some details of the events are listed below:

- The Maule earthquake (*Mw* 8.8) occurred on 27 February 2010. The source was retrieved from Moreno et al. [33]. The largest run-up was registered in Constitución (29 m) [34].
- The Illapel event (*Mw* 8.3) took place on 16 September 2015. The source model was taken from Shrivastava et al. [35]. A large run-up of 10.75 m for this event was recorded, e.g., in Totoral (Coquimbo Region) [36].
- The Valparaíso event happened on 8 July 1730, triggering a tsunami that affected the Chilean coast and was also recorded along Japanese coasts [37]. Large wave heights for this event were registered in Valparaíso (e.g., 9–11 m) [38]. The source model was obtained from Carvajal et al. [38] who suggested that the earthquake size was in the range of *Mw* 9.1–9.3.

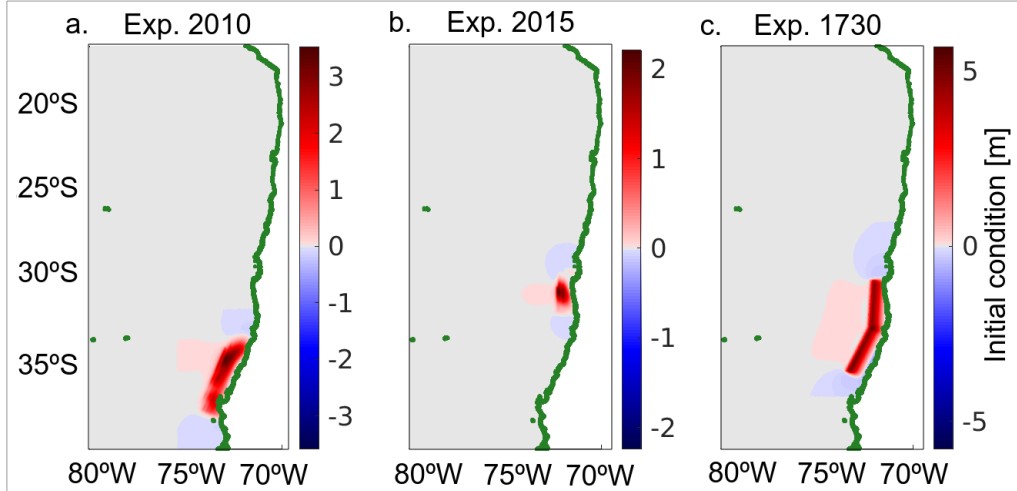

**Figure 3.** Initial sea surface elevation for the three experiments: (**a**). Experiment based on the 2010 Maule earthquake [33]. (**b**). Experiment based on the 2015 Illapel earthquake [35]. (**c**). Experiment based on the 1730 Valparaíso earthquake [38]. More information about slip distribution is shown in Figure S2 of the Supplementary Materials.

While these scenarios and the real data are used as reference to show the behaviour of the three codes, it is worth noting that we do not include any source uncertainties here; rather, we focus only on the outcomes of the comparison of the numerical simulations.

The surface deformation for each model was obtained with the Okada analytical solutions [39]. Subsequently, all models were initialized with identical sea surface perturbations that were bi-linearly interpolated to the triangular mesh, while the initial velocity fields were set to zero. The tsunami propagation was then calculated for three hours, and all models outputs were analysed in identical locations. The processing of inundation areas was facilitated by interpolating the triangular mesh output of TsunAWI to the grid resolution of the finest grids used in Tsunami-HySEA and COMCOT before the comparisons.

### 2.4. Post-Processing the Outcomes

The wave propagation was investigated in virtual and real tide gauge positions. We compared the time series of the sea surface elevation in selected forecast points, as shown in Figure 1. As a measure of consistency between the models, we calculated the correlation coefficients for the first three hours of simulation time. For precise comparisons, all model results were interpolated to the same time instances (30 s resolution), and the coefficients were determined with the MATLAB function "corrcoef". For all tide gauge studies, we used the results for a "reference" Manning value of $n = 0.025$. We covered a large range of depth values, from the tide gauges on the coastline to deep ocean buoys of the DART network. The Supplementary Materials contains additional results.

In the case of the tsunami events with available tide gauge records, the comparison of the experiments was extended to the investigation of model-data consistency. In this regard, we note that we did not aim at achieving a seismic source model validation for the given cases with respect to the observed inundation properties; better agreement with data could be obtained with source optimization and specific parameter tuning [40,41]. Instead, we studied the inter-model's outcome consistency for a fixed source model and did not expect a perfect match with the data. Nevertheless, in the following subsections, we will show the results for each experiment and the comparisons at the offshore and inland gauges, as well as the inundation mapping with their corresponding correlations within the three numerical codes and with the use of different Manning values.

In the case of inundation, run-up properties were investigated for the participating models in a range of Manning values ($n$: 0.015, 0.02, 0.025, 0.035, 0.045, 0.06). We compared inundation extent as well as flow depth and time series at selected forecast points.

We determined the inundation area based on a threshold of 1 cm in all the models for the three experiments, which will be described in the following section. Alternative flow depth thresholds of 1 m and 2 m are shown in the Supplementary Materials.

Details of the simulations and the data used to assess the outcomes of each of the three experiment are shown in Table 2.

**Table 2.** Simulation overview for the three experiments. Abbreviations for the available data used in the study are tide gauge records (TGR), inundation extent (InExt), and flow depth (FLD).

| Exp. ID | Event | Magnitude | Used Data | Comparisons |
|---------|-------|-----------|-----------|-------------|
| 2010 | Maule | *Mw* 8.8 | TGR | Virtual tide gauges (Vtg) and real tide gauges |
| 2015 | Illapel | *Mw* 8.3 | TGR, InExt, FLD | Vtg and real tide gauges InExt for varying Manning $n$ Comparisons to field obs. |
| 1730 | Valparaíso | *Mw* 9.1 | – | Virtual tide gauges InExt for varying Manning $n$ |

## 3. Results

The main focus of this comparison is on the differences with respect to wave propagation and inundation properties. In the following sub-sections, we will summarize comparisons of the time series, inundation extent, and flow depth for the three models as well as various values for bottom roughness controlled by the Manning *n*-value in the coastal area. On the contrary, large differences are not expected in the deep ocean since the linear shallow-water theory delivers a good approximation there; however, differences in resolution might lead to small discrepancies even in the deep sea. The run-up process, on the other hand, is highly non-linear, and different implementations are expected to result in noticeable differences between the models.

For a systematic comparison, all results are regarded in the same regular grid, and the TsunAWI model output is interpolated bi-linearly into the raster positions of Grid 4, which is used in Tsunami-HySEA and COMCOT. Afterwards, inundation values below 1 cm are discarded, and a number of analysis quantities are determined in the three areas of Valparaíso, Viña del Mar, and Coquimbo:

- Inundation area in km$^2$;
- Estimate of inundation volume (integral of max. flow depth) in Mio. m$^3$;
- Maximum run-up height;
- Location of maximum run-up height;
- Mean/max/standard deviation of inundation depth;
- Additional percentiles (e.g., median, 90%, 75%, refer to Tables S1 and S2).

### 3.1. Maule Tsunami 2010

Offshore Assessment Based on Experiment 2010

The source model used to simulate this event is shown in the left panel of Figure 3a. It is based on the seismic source proposed by Moreno et al. [33]. For an evaluation of the wave propagation, we show the time series results at selected tide gauges for three numerical models and compared them to real data when available (Figure 4a–d). Good agreement is visible mostly in the first 90 min. Afterwards, the waveform contains components resulting from reflections on the coast, therefore causing differences to grow. Nevertheless, the correlation between the models in the first three hours is quite high, as can be seen in Table 3 and Figure 4e, which summarise the correlation coefficients of all model-pairs in the first three hours of simulation at all gauge positions in the ocean. The slightly lower correlations at the tide gauge positions "sano", "pich", "sanf", and "talc" might be due to the fact that these stations (tide gauge) are located close to the coast in the coarser grids, thus leading to discrepancies. On the other hand, DART buoy positions show good agreement despite the large difference in resolution between the nested grid (925 m) and the triangular mesh (up to 12 km).

**Table 3.** Correlation coefficients of the time series in Figure 4 for Experiment 2010. The following abbreviations are used—HS: HySEA; TSW: TsunAWI; CC: COMCOT. Refer to Figure 1 for the tide gauge and DART locations.

|       | HS    | TSW   | CC    | HS    | TSW   | CC    |
|-------|-------|-------|-------|-------|-------|-------|
|       |       | **DART 32402** |  |       | **coqu** |  |
| HS    | 1     | 0.977 | 0.973 | 1     | 0.974 | 0.916 |
| TSW   | 0.977 | 1     | 0.967 | 0.974 | 1     | 0.904 |
| Data  | –     | –     | –     | 0.706 | 0.700 | 0.788 |
|       |       | **valp** |  |       | **ts10** |  |
| HS    | 1     | 0.949 | 0.913 | 1     | 0.934 | 0.770 |
| TSW   | 0.949 | 1     | 0.895 | 0.934 | 1     | 0.874 |
| Data  | 0.564 | 0.520 | 0.499 | –     | –     | –     |

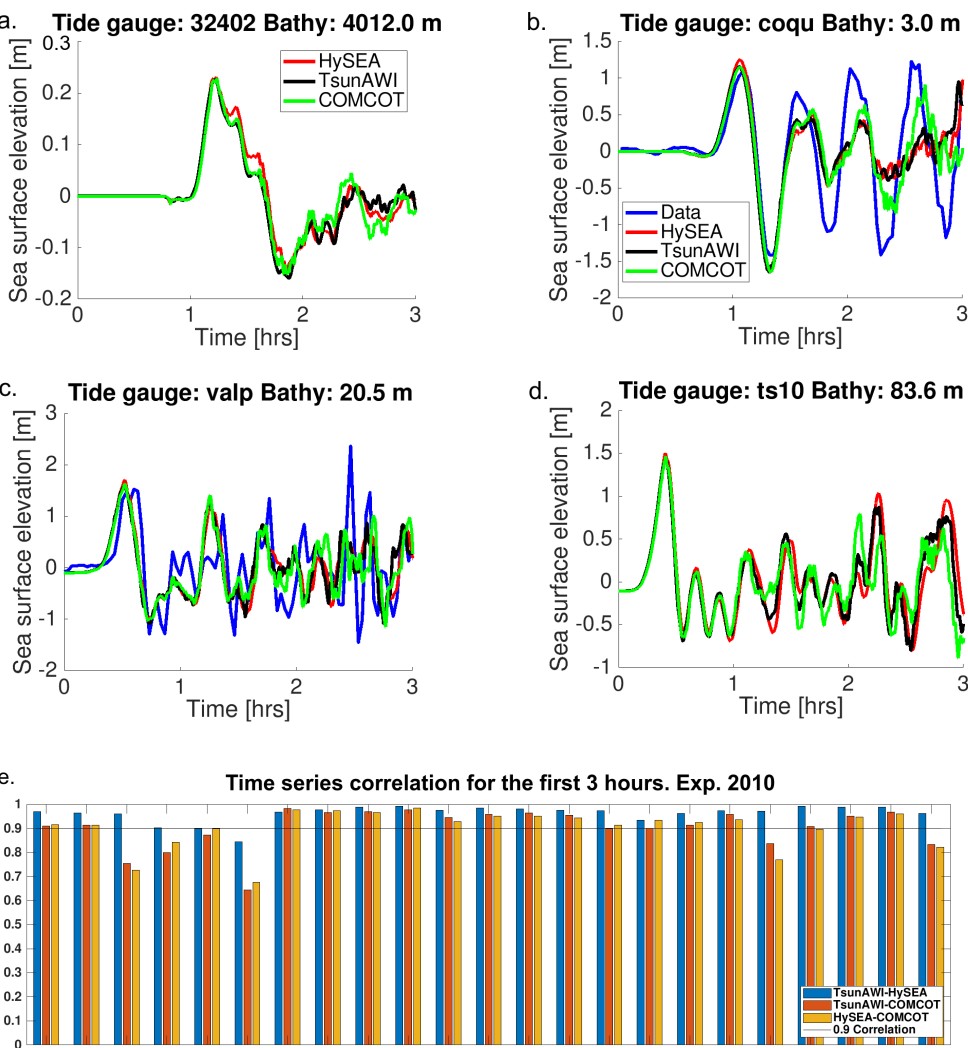

**Figure 4.** Time series and correlations obtained for all models in Experiment 2010 at selected forecast points. Results shown only for Manning $n = 0.025$. (**a**). Time series results for the DART 32402. (**b**). Time series results for the "coqu" tide gauge. (**c**). Time series results for the "valp" tide gauge. (**d**). Time series results for the "ts10" tide gauge. (**e**). Summary of correlations for Experiment 2010 based on a Manning value of $n = 0.025$. The corresponding correlation values are listed in Table 3.

The correlations between the numerical models and the real data are not as high as the consistency between model outcomes. However, it can be seen that at least all codes reproduce the first wave crest well in the tide gauge records of Coquimbo (coqu) and Valparaíso (valp). Better agreement at later stages would require optimizations with respect to the source and better representation of the bathymetric structure in the vicinity of the tide gauge locations.

Note that since only a small inundation occurred in our study areas, hence, we restricted our comparisons to offshore locations.

*3.2. Illapel Event 2015*

3.2.1. Offshore Assessment Based on Experiment 2015

The source model used to simulate this event is shown in the central panel (Figure 3b). The resulting time series at offshore positions agree well within the three numerical models, as can be seen in Figure 5 and Table 4.

Table 4 summarizes the time series where the best correlation was found, specifically for the three tide gauges (coqu, valp, talc) and DART (32402), shown in Figure 5a–d. Even at

the Talcahuano (talc) tide gauge, which is located in the coarser grid (Grid 1) of the nested topo-bathymetry data (Figure 1), correlation values above 0.9 were obtained. Correlations with respect to data are again relatively lower; however, the general features of the observed waves were captured rather well by the models. Figure 5d summarises the correlation at all offshore stations. As in the previous experiments, values above 0.9 were obtained almost everywhere, with few exceptions. However, the exceptions in the deep ocean ("ts7", "ts8", "ts10") do not agree with those in other events, which suggests that a combination of local effects and details of the sources are responsible for the diverging behaviours of the models. Anew, correlations with real data, as shown in Table 4, are lower than between models, and all models particularly underestimate the maximum wave height at the Coquimbo (coqu) tide gauge (Figure 5b). Nevertheless, the general characteristics of the recordings are well-reproduced.

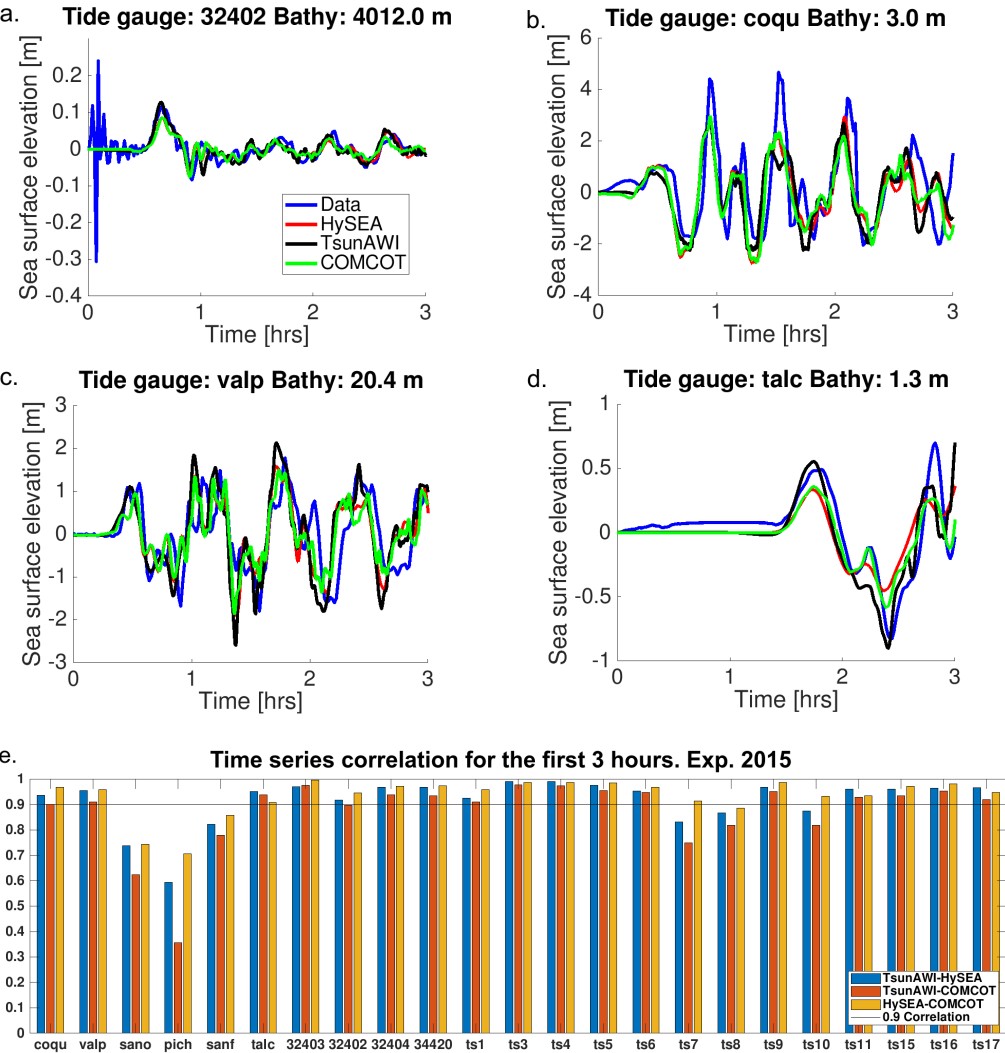

**Figure 5.** Time series obtained for all models in Experiment 2015, shown for selected forecast points. Results are shown only for Manning $n = 0.025$. (**a**). Time series results for the DART 32402. (**b**). Time series results for the "coqu" tide gauge. (**c**). Time series results for the "valp" tide gauge. (**d**). Time series results for the "talc" tide gauge. (**e**). Summary of correlations for Experiment 2015 based on a Manning value of $n = 0.025$. The corresponding correlation values are listed in Table 4.

**Table 4.** Correlation coefficients of the time series in Figure 5 for Experiment 2015. The following abbreviations are used—HS: HySEA; TSW: TsunAWI; CC: COMCOT. Refer to Figure 1 for the tide gauge and DART locations.

| | **HS** | **TSW** | **CC** | **HS** | **TSW** | **CC** |
|---|---|---|---|---|---|---|
| | | **DART 32402** | | | **coqu** | |
| HS | 1 | 0.918 | 0.945 | 1 | 0.936 | 0.967 |
| TSW | 0.918 | 1 | 0.897 | 0.936 | 1 | 0.900 |
| Data | 0.818 | 0.811 | 0.818 | 0.739 | 0.629 | 0.743 |
| | | valp | | | talc | |
| HS | 1 | 0.954 | 0.959 | 1 | 0.950 | 0.907 |
| TSW | 0.954 | 1 | 0.910 | 0.941 | 1 | 0.950 |
| Data | 0.590 | 0.572 | 0.621 | 0.759 | 0.855 | 0.933 |

### 3.2.2. Inundation in Coquimbo Area Based on Experiment 2015

The maximum flow depth obtained by the models for the reference Manning value in a fraction of the Coquimbo area is shown in Figure 6 together with the results in a transect close to that used in Aránguiz et al. [36]. The impact of the tsunami in Coquimbo bay was examined in field surveys (for example, see [36,42]). From these investigations, the measurements obtained are valuable as reference values for comparisons to numerical simulations.

The profile line location is shown in magenta in Figure 6a,c. In Figure 6b,d, we add real data observations that are available along the vicinity of this profile line, taken from Aránguiz et al. [36]. Figure 6b shows the results of the three numerical models based on an *n*-value of 0.025. The large underestimation by the numerical models corresponds quite well with the discrepancy at the Coquimbo (coqu) tide gauge location in Figure 5b, indicating an underestimation of about 2 m by the numerical models. Topography isolines and inundation in the section for the extreme Manning values of $n = 0.06$ and $n = 0.015$ are shown in Figure 6c. At this particular location, the inundation process is affected by steep obstacles in the topographic data set, such as the railway track visible in the central portion of the profile in Figure 6d. This particular configuration and a highly variable bottom relief in general result in an augmented sensitivity with respect to the bottom friction parameter. It becomes clear that the barrier forms an obstacle that may alter the inundation area considerably, which is also visible in the large reduction of the inundation area (more than 64%) in Table 5.

**Table 5.** Inundation areas obtained for the different Manning values in Experiment 2015. The relative drop in estimates refer to the difference between the largest and the smallest values. Inside the brackets, the median values for inundation are shown (in metres). In addition to the median values, other statistical parameters are shown in Table S1 in the Supplementary Materials.

| Model | HS | TSW | CC |
|---|---|---|---|
| **Experiment** | | **2015** | |
| **Location** | | **Coquimbo** | |
| Manning n | | Area (km$^2$) \| Median of flow depth (m) | |
| 0.015 | 2.251 (0.83) | 2.858 (0.92) | 2.503 (0.85) |
| 0.020 | 2.008 (0.85) | 2.527 (0.78) | 2.270 (0.83) |
| 0.025 | 1.735 (0.89) | 2.401 (0.70) | 1.986 (0.87) |
| 0.035 | 1.323 (1.07) | 1.981 (0.68) | 1.491 (0.99) |
| 0.045 | 1.059 (0.99) | 1.453 (0.83) | 1.124 (0.96) |
| 0.060 | 0.769 (0.70) | 1.011 (0.73) | 0.820 (0.65) |
| rel. drop % | 65.8 | 64.6 | 67.2 |

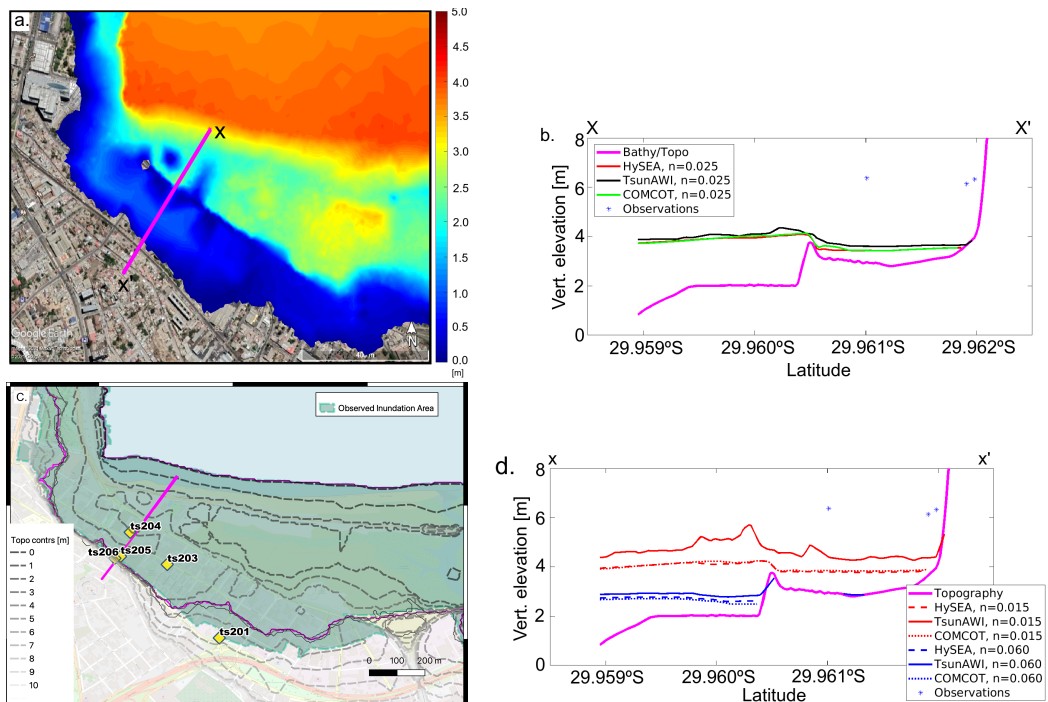

**Figure 6.** (**a**). Maximum flow depths in Coquimbo based on the Experiment 2015 obtained by the model TsunAWI (**top left**) for a Manning value of *n* = 0.025. Refer to Figure S3 in the Supplementary Materials for HS and CC results. (**b**). Maximum flow depth values following the cross section (magenta) shown in the left panels. (**c**). Topography contours over the inundation map in Coquimbo. (**d**). Cross-section showing flow depths that resulted from the three numerical models and Manning values of *n* 0.015 and *n* 0.060. Observation from the 2015 tsunami event in Coquimbo based on Aránguiz et al. [36]. Basemaps in (**a**,**c**) are obtained from © Google Earth 2021, Maxar Technologies © OpenStreetMap Contributors, respectively.

Moreover, we compare the results between the three numerical models for a single Manning value and to the observed inundation area. Here, the inundation extent that resulted from the codes is shown in each panel with the specific Manning value used (Figure 7a–f). Figure 7e,f highlights the main differences derived from this analysis, observed for the largest *n* values.

Alternatively, we also analyse the difference with respect to the Manning values for each model (Figure 8). In this comparison, we highlight the different outcomes of one numerical code with the full range of Manning values, as shown in each panel. These Figures and the relationship between inundation area/volume and Manning *n* in Figure 9a,b show the strong dependency of inundation properties in this particular location. The quadratic regression determined with the MATLAB function "polyfit" describes the inundation relation to be rather consistent, especially with respect to the inundation volume. This is further explored in Figure S4 (Supplementary Materials), where the inundation area is shown for three different flow depth thresholds. Additionally, for higher threshold values (1 m and 2 m), large reduction rates are observed, as shown in the top panel of Figure S4, whereas the lower panel depicts the inundation area in Coquimbo for *n* = 0.025 and three threshold values (1 cm, 1 m, and 2 m).

Table 5 summarizes the values based on Figure 9a,b with the addition of the median value resulting from the flow depth in the whole inundation area. More statistics are listed in the Supplementary Materials. Here, we do not observe a clear functional relationship of the flow depth quantities with respect to the Manning value.

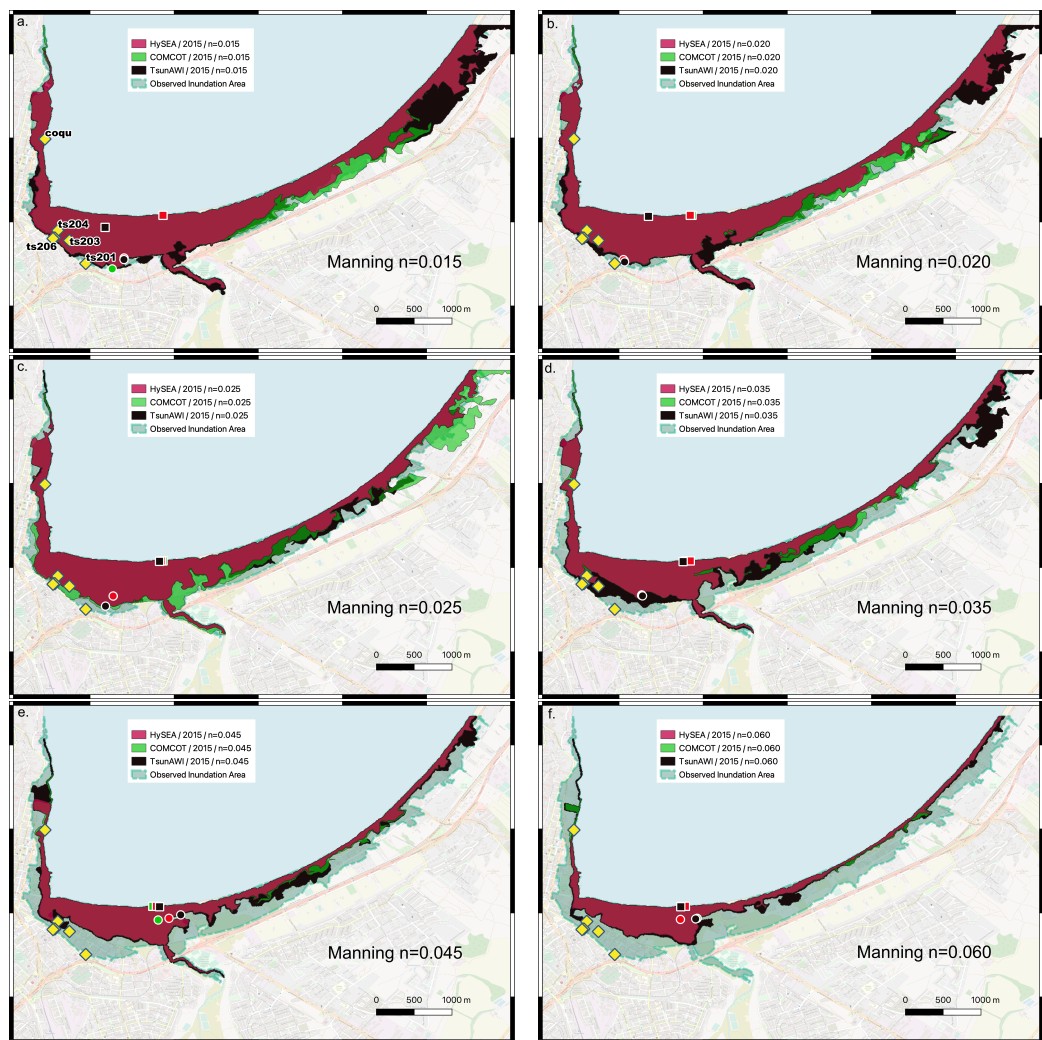

**Figure 7.** Comparison of the inundation areas in Coquimbo resulting from three numerical codes with different Manning values obtained based on the Experiment 2015. (**a**). Manning *n* value of 0.015. (**b**). Manning *n* value of 0.020. (**c**). Manning *n* value of 0.025. (**d**). Manning *n* value of 0.030. (**e**). Manning *n* value of 0.045. (**f**). Manning *n* value of 0.060. Yellow diamonds are virtual tide (coqu) and inland gauges, squares (red, black, green) show positions of the maximum inundation, while circles (red, black, green) show positions of the maximum run-up. Refer to Table S1 (in Supplementary Materials) where these values are summarized. The observed inundation area was taken from SERNAGEOMIN [43]. Basemap: © OpenStreetMap Contributors.

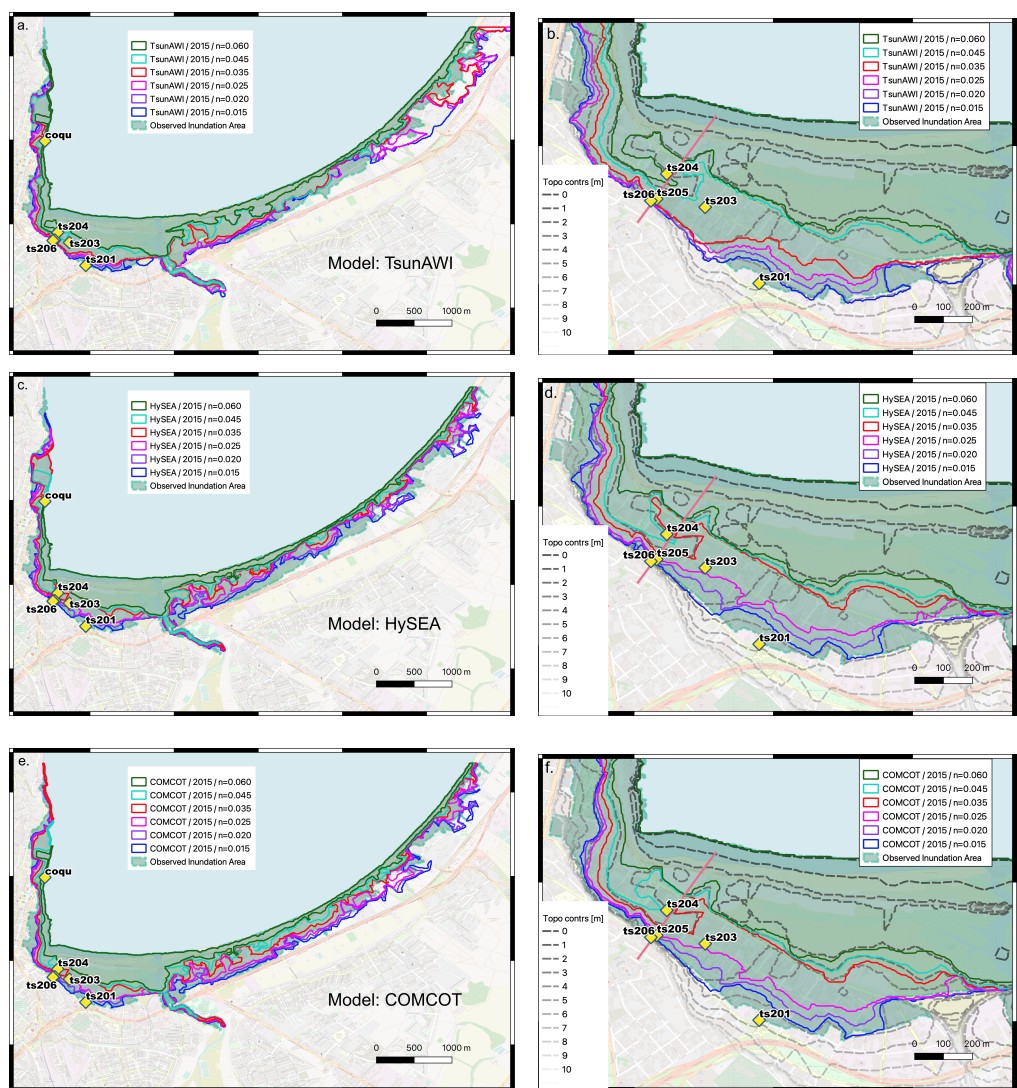

**Figure 8.** Comparison within each numerical model, testing different Manning *n* values. Boundaries showing the extent of inundation areas obtained by the numerical models for all Manning values for Experiment 2015 are shown in each panel: (**a**). All *n* values tested with the TsunAWI model for Coquimbo bay. (**b**). All *n* values tested with the TsunAWI model for southern Coquimbo bay (zoomed area). (**c**). All *n* values tested with the Tsunami-HySEA model along Coquimbo bay. (**d**). All *n* values tested with the Tsunami-HySEA model for southern Coquimbo bay (zoomed area). (**e**). All *n* values tested with the COMCOT model for Coquimbo bay. (**f**). All *n* values tested with the COMCOT model for southern Coquimbo bay (zoomed area). Yellow diamonds stand for virtual inland and tide gauges. Observed inundation area taken from SERNAGEOMIN [43]. Basemap: © OpenStreetMap Contributors.

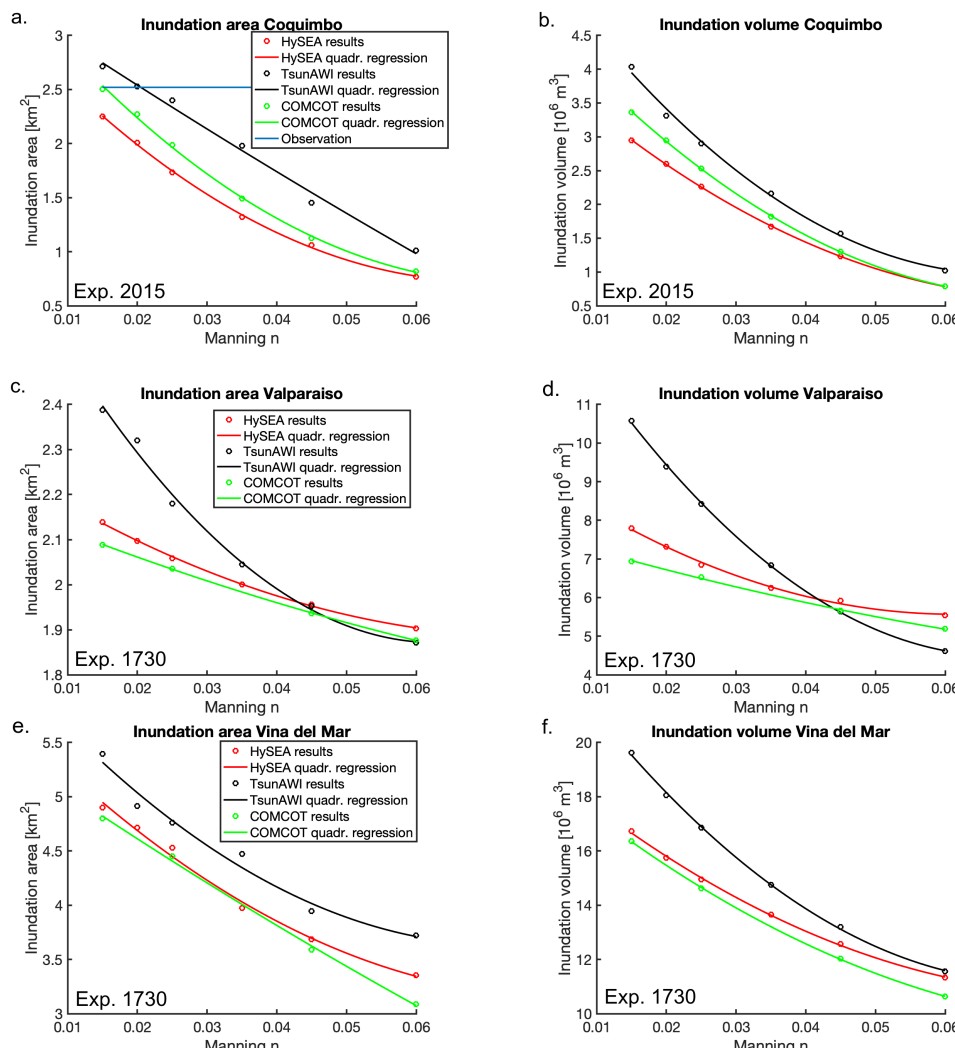

**Figure 9.** Comparisons of the areas and volumes that resulted from the three numerical models based on Experiments 2015 and 1730. (**a**). Inundation area in Coquimbo from Experiment 2015. (**b**). Volume estimates obtained by integrating the maximum flow depth for Experiment 2015. (**c**). Model results for the inundation area in Experiment 1730 in Valparaíso. (**d**). Volume estimates for Experiment 1730 in Valparaíso. (**e**). Model results for the inundation area in Experiment 1730 in Viña del Mar. (**f**). Volume estimates for Experiment 1730 in Viña del Mar, obtained by integrating the maximum flow depth. The lines are quadratic regressions obtained for least squares fit.

### 3.3. Valparaíso Tsunami 1730

#### 3.3.1. Offshore Assessment Based on Experiment 1730

The source of the event is displayed in the right panel of Figure 3c. The initial elevation is based on one of the seismic source models proposed by Carvajal et al. [38], with moment magnitude of *Mw* 9.1. Since no tide gauge observations are available for this event, we compared the consistency of the models with the complete depth range. Due to numerical instabilities that occurred in this configuration of COMCOT after several hours of integration, we restricted our analysis to two hours of simulation (only for this case, along the study we used three hours).

The Figure S5a–d contains the results at selected forecast points, whereas Figure S5e contains a correlation overview of all stations in the ocean and for all model pairs as a bar plot. The correlation values are generally high for any model comparison, with only a few exceptions that are comparable to the situations in the other events. Similar to

Experiment 2015, there are large correlations in the offshore tide gauges. Table 6 lists these corresponding correlation values.

**Table 6.** Correlation coefficients of Experiment 1730 for the first two hours of the time series, also shown in Figure S5 of the Supplementary Materials. Abbreviations used are the following—HS: Tsunami-HySEA; TSW: TsunAWI; CC: COMCOT. Refer to Figure 1 for the tide gauge and DART locations.

|  | **HS** | **TSW** | **CC** | **HS** | **TSW** | **CC** |
|---|---|---|---|---|---|---|
|  | | **DART32402** | | | **valp** | |
| HS | 1 | 0.979 | 0.984 | 1 | 0.975 | 0.935 |
| TSW | 0.979 | 1 | 0.982 | 0.975 | 1 | 0.946 |
|  | | ts3 | | | ts8 | |
| HS | 1 | 0.982 | 0.966 | 1 | 0.955 | 0.772 |
| TSW | 0.982 | 1 | 0.972 | 0.955 | 1 | 0.805 |

### 3.3.2. Inundation in Valparaíso and Viña del Mar Based on Experiment 1730

The resulting inundation depth distribution for the three models in Valparaíso and Viña del Mar is shown in Figure 10a–c, which shows a maximum flow depth of 8 m in the city of Viña del Mar. In particular, Figure 10d shows the topography profile in magenta as a reference to the flow depth values retrieved along this profile. Main variations of a few centimetres are given by the COMCOT code (green line) for this particular Manning $n$-value of 0.025.

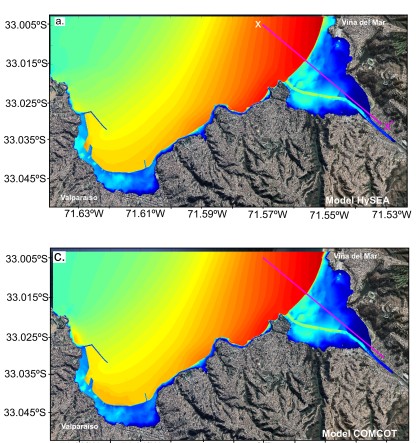
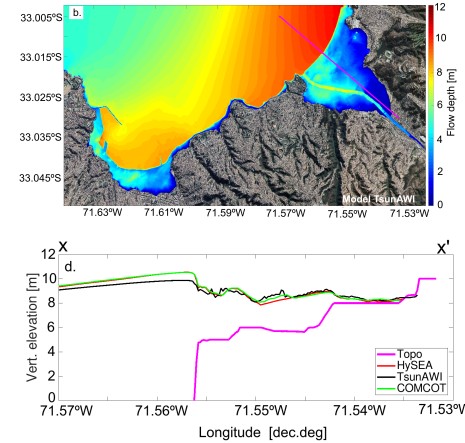

**Figure 10.** Maximum flow depths in Valparaíso and Viña del Mar obtained by the three numerical models for a Manning value of $n = 0.025$ in Experiment 1730. (**a**). Results from Tsunami-HySEA, (**b**). TsunAWI, (**c**). COMCOT. The magenta line stands for the extent of the cross section. (**d**). Lines show the maximum wave amplitude (relative to shoreline) resulting from each numerical model and the elevation above sea level (Topo). Basemap: © Google Earth 2018.

Furthermore, the differences between models and for varying Manning values are presented in Figure 11. The topographical structures of the two domains differ considerably. In Valparaíso, a large part of the coastal zone consists of harbour constructions, and a rather narrow strip of low topography is bounded by steep hills. On the other hand, the topographical structure of Viña del Mar is smoother, with a large part of the city located close to sea level. Topography isolines of both locations are included in Figure 11. As a result from the distinct settings, the sensitivity with respect to the bottom friction differs, as can be seen in Figure 11 and Table 7. Whereas the inundation area in Valparaíso drops over the whole Manning range by a factor of about 10% for the nested grid models and 21.6% for

TsunAWI, this reduction is considerably larger (>30%) in Viña del Mar, where the inundated area stretches much further inland, and bottom friction acts over a larger distance.

The inundation process is highly non-linear, and small differences in the discretization of the topography may result in large differences in the run-up simulation. COMCOT and Tsunami-HySEA operate on identical meshes, and the results are quite similar, whereas the results of TsunAWI differ more clearly, as exemplified in Figure 12, which displays the inundation area in Viña del Mar for all models and Manning values. TsunAWI yields an overestimation of the inundation area as compared to the other codes in many of the cases (Figure 12a,d).

The dependency of the inundation areas and the corresponding estimates of volume derived from the integral of the maximum flow depth in the affected area is shown in Figure 9c–f for Valparaíso and Viña del Mar. We include quadratic regression polynomials and get very good agreement, especially with respect to the volume (Figure 9d,f). However, the coefficients depend on the location and the model. As mentioned before, the terrain shape influences the run-up characteristics, and consequently the curves are steeper in the case of Viña del Mar. In that respect, Figure 9 shows that TSW in general yields a steeper behavior when areas and volumes are compared, which is probably related to the extrapolation method applied in that numerical model.

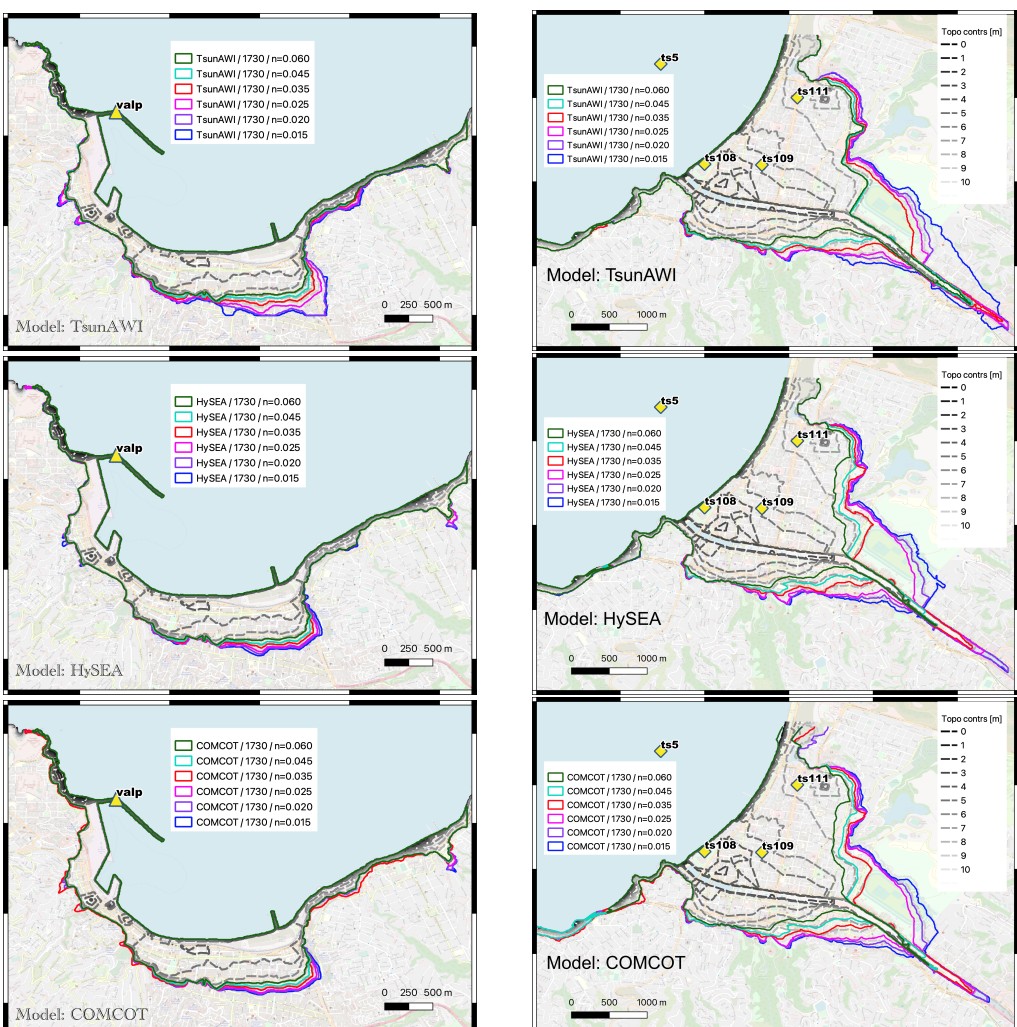

**Figure 11.** Extent of inundation areas obtained by the three numerical models for all Manning values tested in Experiment 1730 in Valparaíso (**left panels**) and Viña del Mar (**right panels**). Yellow diamonds show the virtual tide (valp) and inland gauges. Basemap: © OpenStreetMap Contributors.

Another question of interest is the influence of mesh resolution and thus the representation of topography in the model. One of the compelling results from the work by Griffin et al. [24] is that a mesh resolution of 25 m is suitable for estimations of the inundation extent, and that otherwise, data quality is more important than the mesh resolution. We investigated the influence of mesh resolution in the model TsunAWI and compared the results for different resolutions of the triangular mesh in the pilot areas. To this end, by conducting experiments with a Manning value of $n = 0.025$, in meshes with five different resolutions ranging from a mean resolution of approximately 100 m (finest resolution: 12.6 m) to a mean edge length of about 10 m (finest resolution: 2.5 m), we arrived to a similar conclusion. For the experiments conducted here, we find that the sensitivity regarding $n$ is generally larger than the sensitivity to the mesh resolution.

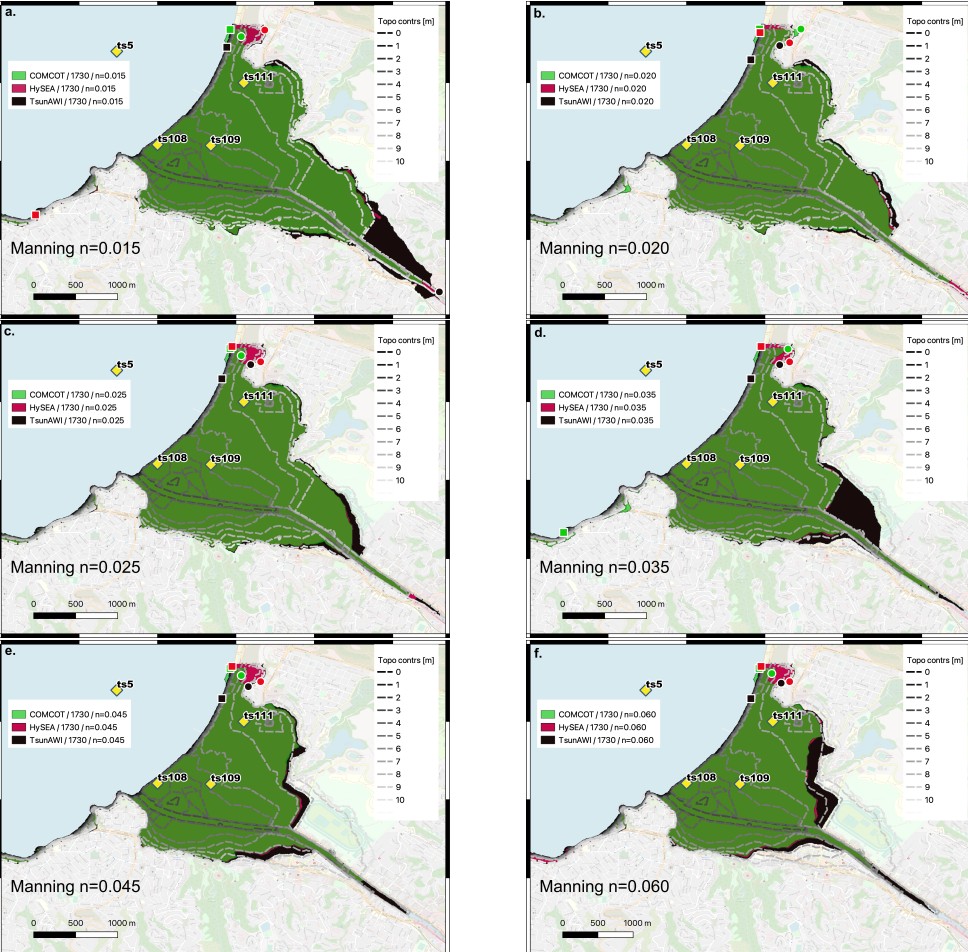

**Figure 12.** Inundation area in Viña del Mar for all Manning values obtained by the three models for Experiment 1730. (**a**). Manning *n*-value of 0.015. (**b**). Manning *n*-value of 0.020. (**c**). Manning *n*-value of 0.025. (**d**). Manning *n*-value of 0.030. (**e**). Manning *n*-value of 0.045. (**f**). Manning *n*-value of 0.060. Yellow diamonds are virtual tide (ts5) and inland gauges, squares (red, black, green) show positions of the maximum inundation, while circles (red, black, green) show positions of the maximum run-up. Basemap: © OpenStreetMap Contributors.

Figure 13a shows the mesh structure of the coarsest and finest mesh in the port area of Valparaíso. The resolution in the offshore part is controlled by the CFL criterium and is comparable in both meshes, whereas the coastal land part differs strongly. Figure 13b,c shows the inundation areas in Viña del Mar, Valparaíso, and Coquimbo for all meshes. Only small variations are visible. Table 8 summarises the inundation areas for all experiments and meshes. The spread between the largest and the smallest values is below 10%, and

although there is no consistent relationship between resolution and value, the coarsest mesh generally shows an overestimation when compared to the finer resolutions.

The temporal evolution of the inundation process is shown in Figure 14 for two locations in Viña del Mar. One position is very close to the coastline (ts108), the second one is further inland (ts109: at a distance of about 500 m from the shoreline and 600 m away from ts108; see Figure 12). As can be seen in Figure 14a,b, the models agree well in arrival time and the general shape of the time series in these locations, while the first crest yielded the highest flow depth. In the model comparison for a Manning value of *n* = 0.025, the maximum value ranges from 4.51 to 4.96 m. The drying process differs for the models since non-identical algorithms were used; the extrapolation scheme in TsunAWI shows more fluctuations, especially for small Manning values.

**Table 7.** Inundation area obtained in Experiment 1730, in the domains of Valparaíso and Viña del Mar, for all models and ranges of Manning values tested in this study. Relative drop shows the decrease between the largest and the smallest area values. The median relates to the inundation estimates for the entire area. Besides the median values, other statistical parameters are shown in Tables S2 and S3 in the Supplementary Materials. Refer to Figures 9c–f and 11.

| Model | HS | TSW | CC | HS | TSW | CC |
|---|---|---|---|---|---|---|
| Experiment | | 1730 | | | 1730 | |
| Location | | Valparaíso | | | Viña del Mar | |
| Manning n | Area (km$^2$) | Median of flow depth (m) | | Area (km$^2$) | Median of flow depth (m) | |
| 0.015 | 2.139 (3.50) | 2.388 (4.23) | 2.089 (3.02) | 4.900 (2.71) | 5.396 (2.94) | 4.800 (2.69) |
| 0.020 | 2.097 (3.25) | 2.320 (3.83) | 2.070 (3.0) | 4.713 (2.70) | 4.914 (3.01) | 4.715 (2.77) |
| 0.025 | 2.059 (3.05) | 2.180 (3.6) | 2.036 (2.90) | 4.529 (2.67) | 4.761 (2.93) | 4.450 (2.66) |
| 0.035 | 2.001 (2.87) | 2.045 (3.10) | 2.085 (3.14) | 3.974 (2.83) | 4.473 (2.69) | 3.930 (2.83) |
| 0.045 | 1.956 (2.77) | 1.952 (2.63) | 1.936 (2.68) | 3.687 (2.78) | 3.946 (2.72) | 3.591 (2.69) |
| 0.060 | 1.903 (2.72) | 1.871 (2.23) | 1.876 (2.56) | 3.355 (2.71) | 3.720 (2.45) | 3.087 (2.58) |
| rel. drop % | 11.0 | 21.6 | 10.2 | 31.5 | 31.1 | 35.7 |

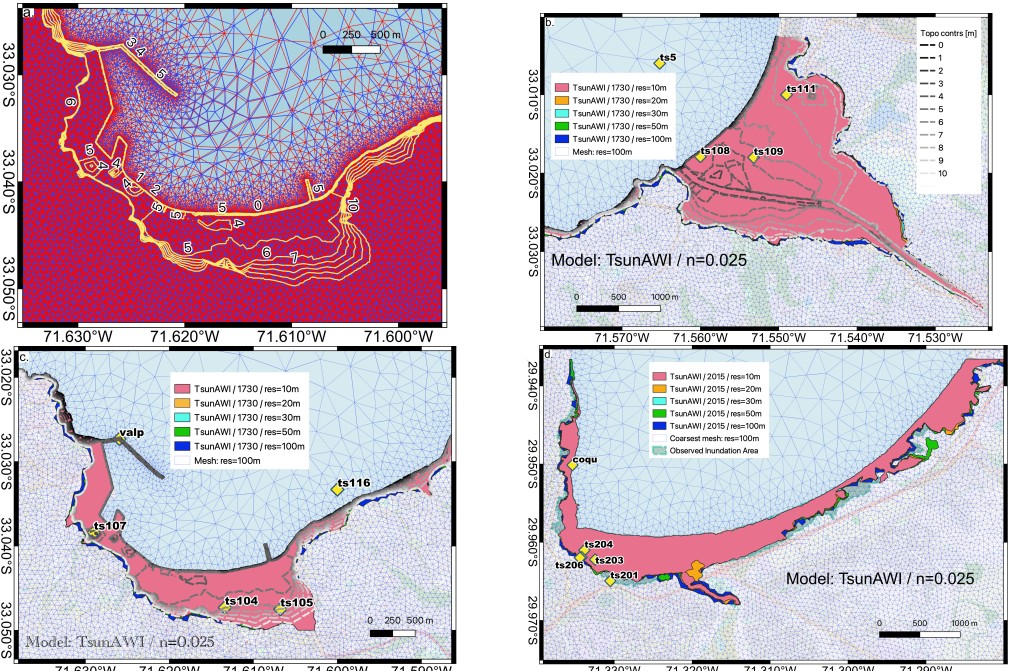

**Figure 13.** Comparison of the inundation dependency on bathymetry. (**a**). Coarsest mesh (shown in blue) and finest triangulation (shown in red) in the Valparaíso area. (**b**). Inundation area in Viña del Mar in all meshes for Experiment 1730 and a Manning value of *n* = 0.025, calculated with TsunAWI. (**c**). Inundation area in Valparaíso in all meshes for Experiment 1730 and a Manning value of *n* = 0.025, calculated with TsunAWI. (**d**). Inundation area in Coquimbo in all meshes for Experiment 1730 and a Manning value of *n* = 0.025, calculated with TsunAWI. Yellow diamonds show the tide and inland gauges. Basemap: © OpenStreetMap Contributors.

Figure 14c,d shows the time series in the aforementioned locations for the TsunAWI experiments with different Manning values. The leading wave crest at location ts108, right on the coast, increases with the growing roughness, whereas the maximum flow depth further inland drops with rising Manning values due to the continuous effect of bottom friction during the inundation process.

**Table 8.** Inundation area in pilot regions (refer to a minimum flow depth of 1 cm) obtained by TsunAWI in all meshes. The relative reduction is obtained by computing the ratio of the difference between the largest and the smallest value with reference to the maximum.

| Mesh | | Inundation Area (km$^2$) | | |
| --- | --- | --- | --- | --- |
| Mean Res. | Finest Res. (m) | Viña (1730) | Valparaíso (1730) | Coquimbo (2015) |
| 100.0 | 12.7 | 4.955 | 2.392 | 2.503 |
| 50.0 | 12.7 | 4.841 | 2.307 | 2.432 |
| 30.0 | 9.5 | 4.793 | 2.164 | 2.364 |
| 20.0 | 6.1 | 4.761 | 2.180 | 2.401 |
| 10.0 | 2.5 | 4.711 | 2.276 | 2.384 |
| rel. reduction (%) | | 4.8 | 9.6 | 5.6 |

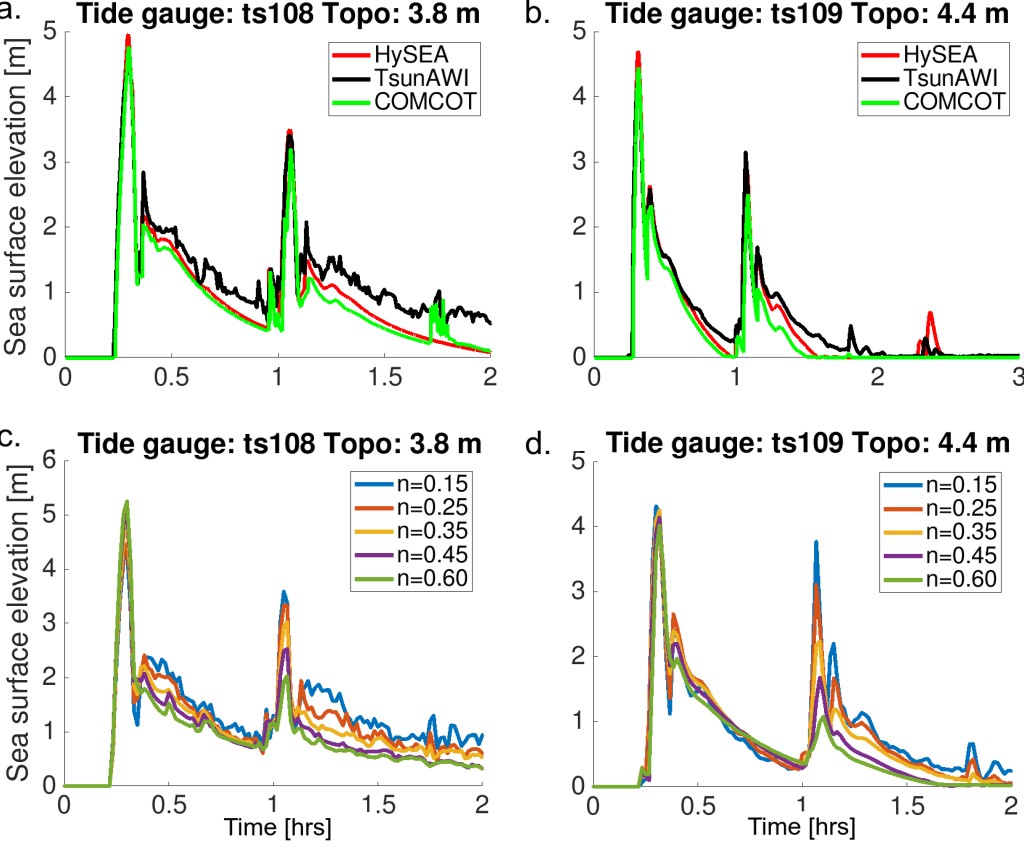

**Figure 14.** Upper panels: Numerical code comparison for Experiment 1730. Temporal evolution of the inundation process at two locations in Viña del Mar: (**a**). Virtual tide gauge "ts108" close to the coast. (**b**). Virtual tide gauge "ts109" about 500 m inland. Lower panels: Comparison of TsunAWI results for the full range of Manning values for Experiment 1730. Temporal evolution of the inundation process at two locations in Viña del Mar: (**c**). Virtual tide gauge "ts108". (**d**). Virtual tide gauge "ts109" about 500 m inland. Refer to Figure 13 for their locations.

Table 7 summarizes the values shown in the regressions in Figure 9, with the addition of the median value (shown in brackets) that resulted from the flow depths in the

whole inundation area, shown in Figure 11. Here, as in the above-mentioned Experiment 2015, we do not observe a clear functional relationship of the flow depth quantities with respect to the Manning value. More statistics are listed in the Supplementary Materials (Tables S2 and S3).

## 4. Discussion

We conducted a series of numerical experiments with three simulation codes and three seismic scenarios with different magnitudes and wave-lengths allowing a comprehensive comparison about the behaviour of the codes and the effects of the Manning parameters. Herein, we show the differences in wave propagation and inundation as well as in sensitivity with respect to the bottom roughness parameter. From the outcomes of the numerical simulations based on these seismic sources, we found consistent time series within the first 90 min of the model simulations at (offshore) tide gauge locations, and in many cases, even for a longer time range. The correlation coefficients for up to three-hour comparisons were generally high despite large differences in the spatial resolution in the deep ocean. We observed fairly lower correlations at the tide gauges located in the shallow coastal areas of the coarser grids, which might be related to different representations of the local wave evolution (or transition of numerical schemes). In general, the models show similar agreement with data records, although improvements could be obtained by source optimization and better resolution of local bathymetric features.

The sensitivity of inundation properties with respect to bottom friction strongly depends on the local topography. For both the 2015 and 1730 experiments, we analysed the inundation at three locations with distinct relief shapes and obtained strong dependency in all cases. However, the actual relations with respect to inundation area and volume differ considerably. More specifically, we obtained quadratic dependency between the Manning $n$ and inundation volume; however, the coefficients depended strongly on the specific topography characterization. As for the large $Mw$ 9.1 event from 1730, which yielded large coastal wave heights, simulations showed smaller sensitivity for the inundation area and volume with respect to the Manning value in Valparaíso, which has a rather narrow (<800 m), low-lying area bounded by hills. On the contrary, in the case of Viña del Mar, the inundation stretched far inland, and the inundation stage was affected by bottom friction over a large distance. We expect the situation to be different for smaller events with lower coastal wave heights. In contrast, based on the Coquimbo 2015 experiment, the sensitivity of inundation with respect to bottom friction is high due to a strongly variable bottom topography. These differences can be seen in the inundation maps for the entire Coquimbo area and the southern Coquimbo bay (Figure 8), a region prone to tsunami resonance [36].

Moreover, we also observed the strong reduction of the inundation area in the results of Experiment 1730 in Coquimbo. Figure S6 of the Supplementary Materials displays the TsunAWI results for the three Manning values, which shows a reduction of the inundation area over the full range of 52.8%. While most of the results were obtained with respect to a minimum flow depth of 1cm, as alternative reference values, we conducted some of the experiments based on inundation thresholds of 1 m and 2 m. We obtained similar results with respect to the relationship between the Manning $n$ value and the inundation area/volume. Figures S4 and S6 contain the results for TsunAWI in Coquimbo for two experiments (2015 and 1730, respectively). As for the dependency of the inundation results on the numerical model, we found very good agreement between the nested grid models and clearer differences in the case of TsunAWI, especially for low Manning numbers. In that context, it is worth mentioning that numerical details of the implemented inundation scheme together with the mesh topology as well as the type of topography data (terrain or surface model) will also affect the results. Those aspects are beyond the scope of this study and needs further investigation.

The relationship between flow depth and bottom friction depends on the local topographic setting as well. Close to the coast, the joint effect of a stronger resistance and a

steep slope can lead to a higher flow depth for a larger bottom friction, whereas in longer distances from the coast, values drop for larger Manning values due to the accumulated influence over a longer path. Figure 15 exemplifies this relationship in the case of Viña del Mar and Experiment 1730. The picture shows the ratio of maximum flow depth obtained for the extreme Manning values of $n = 0.060$ and $n = 0.015$. In a strip of about 350 m along a section of the coastline, the larger Manning values result in larger maximum flow depth. In the northern part of the model domain, the situation changes, probably due to a smoother bathymetry condition in the near-shore range; however, more investigation is needed. All models show similar behaviour although details differ, as can be seen in varying lines of equal inundation for the two Manning values.

Manning values between $n = 0.02$ and $n = 0.025$, which are commonly used for tsunami hazard assessment in the entire domain, show similar inundation extent within codes, except for the TSW code influence in northern Coquimbo in the inland domain (topography), which is possibly related to the extrapolation scheme in the triangular mesh. It is important to mention the influence of topo-bathymetric data when comparing, for instance, the effects of the 1730 tsunami event in Valparaíso, Viña del Mar, and Coquimbo. Here, topography is very different; these variations within Coquimbo are even more evident in the northern and southern part of the bay. Thus, the cases where strong differences in resolution or bathymetry information exist need further investigations, specially in light of the varying Manning $n$-values at the local scale.

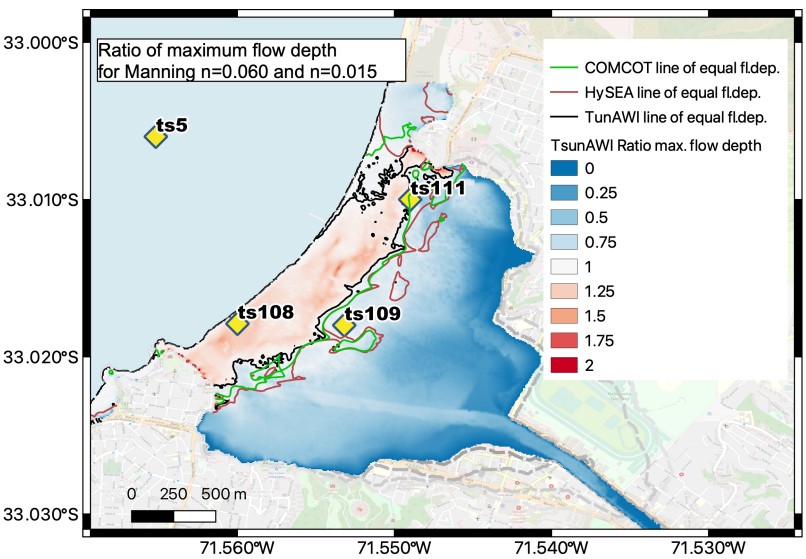

**Figure 15.** Ratio of the maximum flow depth obtained for the extreme Manning values (flow depth ($n = 0.06$)/flow depth ($n = 0.015$)) in the intersection of inundation areas in Viña del Mar. Example that summarizes the outcomes of Experiment 1730. Basemap: © OpenStreetMap Contributors.

## 5. Conclusions

In this study, we specifically investigated the sensitivity and consistency within numerical model results that might be used for tsunami hazard assessment and tsunami warning products. From our simulations using three different numerical approaches, we found consistent estimates for the near-shore wave amplitude of the leading wave crest for identical source models regardless of the different mesh topologies. Furthermore, the onset of signals between the models shows very good agreement; thus, we expect consistent estimates of wave amplitude and arrival time in the near-shore range as long as sources are identical and the bathymetry data are consistent.

While there is a rather clear relationship connecting the total inundation area and volume to the Manning number obtained by all numerical models, albeit with gradual differences, the local inundation process is highly non-linear and very dependent on data and

inundation scheme. Hence, main differences can be seen between the nested grid models and TsunAWI due to the different discretization of topography and inundation processes.

Finally, with respect to inundation, we found that the most outstanding outcome is the high sensitivity of the inundation properties to Manning values. The choice of Manning values leading to differences of 11–65% were seen in the inundation areas. This suggests that considerable attention must be given to the choice of bottom friction parameters in the numerical simulations of tsunami inundation, especially in the preparation of warning products and tsunami risk preparedness plans. The case of spatially varying friction parameters, which could provide much better agreement with land use, needs additional investigations. Thus, future work should consider these uncertainties in tsunami numerical modelling.

**Supplementary Materials:** The following supporting information can be downloaded at: https://www.mdpi.com/article/10.3390/geohazards3020018/s1. Figure S1: Small section of the triangular mesh used in the TsunAWI simulations. Figure S2: Slip distribution for the three seismic sources used as inputs. Figure S3: Maximum flow depths in the Coquimbo area that resulted from two numerical models: Tsunami-HySEA and COMCOT for the Manning value of $n = 0.025$, for Experiment 2015. Figure S4: Inundation area and functional relation for the full Manning range based on Experiment 2015. Figure S5: Time series and correlations obtained for all models in Experiment 1730 for selected forecast points. Figure S6: Inundation area obtained with TsunAWI based on Experiment 1730 in the Coquimbo area. Tables S1–S3: Summary of statistics of the inundation results for each numerical code based on Experiments 2015 and 1730.

**Author Contributions:** Conceptualization, S.H., N.Z., and A.G.; numerical simulations, S.H., N.Z., and A.G.; TsunAWI model development, N.R. and S.H.; data curation, N.Z. and A.G.; writing the original draft, S.H. and N.Z; review and editing of the manuscript, S.H., N.Z., A.G., and N.R.; visualization, S.H. and N.Z.; project administration, N.R.; funding acquisition, N.R. All authors have read and agreed to the published version of the manuscript.

**Funding:** Part of this research was funded by the German Federal Ministry of Education and Research within the project RIESGOS, grant number: 03G0876C.

**Data Availability Statement:** HySEA code information can be accessed at: https://edanya.uma.es/hysea/index.php/models/tsunami-hysea (last accessed 12 December 2021). TsunAWI code and information are available at https://gitlab.awi.de/tsunawi/tsunawi (Last accessed 6 June 2022). GEBCO freely available: https://www.gebco.net/ (Last accessed in 8 August 2021).

**Acknowledgments:** This study is part of the tsunami component in the RIESGOS project, with a larger scope of multi-hazard assessments in the Andes region. The RIESGOS project is funded by the German Federal Ministry of Education and Research (Grant numbers 03G0876C and 03G0905C). AG acknowledges the Research Center for Integrated Disaster Risk Management (CIGIDEN), ANID/FONDAP/15110017. NZ has received funding from the Marie Skłodowska-Curie grant agreement H2020-MSCA-COFUND-2016-75443. The Servicio Hidrográfico y Oceanográfico de la Armada de Chile (SHOA) provided high-resolution bathymetry data via the CENDHOC program (Centro Nacional de Datos Hidrográficos y Oceanográficos de Chile). We thank EDANYA Group for sharing the Tsunami-HYSEA code and X. Wang for sharing the COMCOT code. We acknowledge M. Moreno, M. Shrivastava and M. Carvajal for providing the finite faults used in the three experiments. Basemaps by OpenStreetMaps contributors were used in most of the figures. Basemaps in Figures 6 and 10 are taken from Google Earth. Some figures were generated with the GMT software [44]. TsunAWI code optimization was supported by the LEXIS project, funded by the EU's Horizon 2020 Research and Innovation Programme (2014–2020) under grant agreement no. 825532. Some figures were generated with the software QGIS. We would like to deeply thank the editors and three anonymous reviewers for valuable comments that helped us improving the manuscript.

**Conflicts of Interest:** The authors declare no conflict of interest.

## Abbreviations

The following abbreviations are used in this manuscript:

HS Tsunami-HySEA code or numerical model
CC COMCOT code or numerical model
TSW TsunAWI code or numerical model
FD Tsunami flow depth

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
