# Peer review of "Systematic Comparison of Tsunami Simulations on the Chilean Coast Based on Different Numerical Approaches"

_2624-795X, doi:10.3390/geohazards3020018_

Round 1

Reviewer 1 Report

In the Introduction section, it would be quite useful to incorporate some basic ideas expressed by the recent paper by Behrens, J. et al. 2021. Probabilistic Tsunami Hazard and Risk Analysis: A Review of Research Gaps. Front. Earth Sci. 9:628772. doi: 10.3389/feart.2021.628772.

45 ….set-up (uncertainties) used…Replace by …set-up uncertainties that could be used…

49 ….in selected points of interest (POIs). Replace by …. in selected forecast points.

77  Reference [18] is cited as only “submitted” while no authorship is included. Please add authorship otherwise remove.

110 2. Methods. Replace by 2. Methods and Data.

112 ….of the the numerical. Correct to… of the numerical.

Fig. 1. It would be useful to distinguish by different symbols the real tide gauges, the virtual tide gauges, and the Deep-ocean Assessment and Reporting of Tsunamis (DARTs).

Author Response

Response to Reviewer 1

We greatly appreciate the comments from the Reviewer. Please see the following responses to your comments:

Comments and Suggestions for Authors

In the Introduction section, it would be quite useful to incorporate some basic ideas expressed by the recent paper by Behrens, J. et al. 2021. Probabilistic Tsunami Hazard and Risk Analysis: A Review of Research Gaps. Front. Earth Sci. 9:628772. doi: 10.3389/feart.2021.628772.

  • Response: Thank you for the suggestion, indeed the paper contains a comprehensive overview of research gaps in the tsunami modeling context. We have added the reference to Behrens et al. (2021) in the Introduction of the manuscript. 

45 ….set-up (uncertainties) used…Replace by …set-up uncertainties that could be used… 

  • Response: It has been changed to “Although uncertainties with respect to the tsunami source might pose the largest problem for reliable hazard forecasting, it is still important to estimate uncertainties due to the numerical set-up used to generate the warning products”. Now in Line 43

49 ….in selected points of interest (POIs). Replace by …. in selected forecast points.

  • Response: We use forecast points for general (virtual tide gauges, inland gauges, DART, real tide gauges).

77  Reference [18] is cited as only “submitted” while no authorship is included. Please add authorship otherwise remove.

  • Response: We corrected the mistake in the reference file. Thank you for pointing this out.

110 2. Methods. Replace by 2. Methods and Data. 

  • Response:  We replaced to Data and Methods

112 ….of the the numerical. Correct to… of the numerical.

  • Response: Corrected

Fig. 1. It would be useful to distinguish by different symbols the real tide gauges, the virtual tide gauges, and the Deep-ocean Assessment and Reporting of Tsunamis (DARTs).

  • Response: Now four different colors are used in Figure 1 to highlight the distinction.

Reviewer 2 Report

Overview;

In this paper, the authors showed the comparison of tsunami simulation results in the case that simulation models (resolutions) are changed, and roughness parameters are changed. But the outcomes of this research are only them. Indeed, the things that roughness contributes to inundation area or depth are obvious!! This finding is not novel. For publishing this paper, more scientific contribution should be shown. If the main topic of this research is parameter study of numerical simulation, more parameters should be tested not only roughness/resolution.

And, here is the other comments.

-Line 24-25, 102-104, this should be described in Acknowledgement. This information is not important for readers.

-Table 1, Please explain why the authors used different grid-cell sizes (second domain) for Tsunami-HySEA and COMCOT.

-Line 211-217, fault parameters (e.g., strike, slip, dip) used for the simulations should be shown in Table.

-Figure3. Delete Time 0 sec.

-For instance, in Table. 3, why are the correlation coefficients shown? Please explain the importance of this. For the comparison of model accuracy, correlation coefficient is not suitable. The difference of simulation results from observed values (e.g., geometric average value, variability)is important for model comparison.

Author Response

Reply to Reviewer 2

Comments and Suggestions for Authors

Overview: In this paper, the authors showed the comparison of tsunami simulation results in the case that simulation models (resolutions) are changed, and roughness parameters are changed. But the outcomes of this research are only them. Indeed, the things that roughness contributes to inundation area or depth are obvious!! This finding is not novel. For publishing this paper, more scientific contributions should be shown. If the main topic of this research is parameter study of numerical simulation, more parameters should be tested not only roughness/resolution.

  • We are thankful for the revision and the suggestions the reviewer makes. Our previous version may have lacked some clarity related to the contribution, thus, we have now tried to make it more specific and added that our scope is specifically based on the numerical code, roughness effects on propagation and inundation, and we added comparisons related to the model representation of  topo-bathymetry.

  • Response: Thanks a lot for the helpful suggestions. Indeed, the general relationship between inundation area and bottom friction for a given tsunami source is known. However, to our knowledge the specifics of the dependency between inundation area or flow depth and the roughness parameter or parameterization is less studied and it is exactly our aim to investigate the actual dependency systematically for different models, events and topographical settings.

    For applications like probabilistic inundation studies or early warning the calculated flow depth distribution may be of crucial importance and therefore, we consider it relevant to investigate the influence beforehand.

    To our knowledge it is not common to use detailed roughness maps in inundation studies, frequently single Manning values are used in large model domains and we show that results may vary considerably for different values. In fact, we considered including parameters like time step, or viscosity into the comparison, however due to the differing model implementations it would require to compare the technical details of the codes, which is not our aim. Rather we would like to compare the default setups of the models and concentrate on the most obvious parameters adjusted by the users being the mesh/grid and the Manning parameter. Therefore, we would like to stick to roughness and resolution comparisons in the study (please refer to Figure 13 in the manuscript). Further work in this direction is intended but using other coastal sites. Gibbons et al (2022) has already tackled a wider problem in a probabilistic manner where fault parameters and velocity, as well as roughness were investigated.

And, here are the other comments.

 -Line 24-25, 102-104, this should be described in Acknowledgement. This information is not important for readers. 

  • Response: This has been followed and we moved the entry in the abstract to the Acknowledgement. However, we would like to keep the remark in the Introduction, since we consider it relevant to point out the connection of inundation depth values to downstream results like loss estimates in a multi hazard assessment, which is investigated in the RIESGOS project. This missing explanation was added to the Introduction.

 -Table 1, Please explain why the authors used different grid-cell sizes (second domain) for Tsunami-HySEA and COMCOT.

  • Response: Grid 2 to is the same, thank you for pointing that out. Actually, we use the same extent and resolution. While Comcot uses a xyz text format, for HySEA we just need to convert this to a grd format. Thus, it is the same and we have now corrected the mistake.

 -Line 211-217, fault parameters (e.g., strike, slip, dip) used for the simulations should be shown in Table. 

  • Response: We use finite fault models available at the Earthquake Source Model Database from Mai and Thingbaijam (2014) found at [http://equake-rc.info/srcmod/]. We have added this information in the Supplementary Material, with figures showing slip model and deformation. In the case of the 1730 source parameters, this was obtained personally from Matías Carvajal, from the paper Caravajal et al. (2017) and is shown also in the Supplementary Material. 

 -Figure3. Delete Time 0 sec. 

  • Response: Corrected, now we use deformation.

 -For instance, in Table. 3, why are the correlation coefficients shown? Please explain the importance of this. For the comparison of model accuracy, correlation coefficient is not suitable. The difference of simulation results from observed values (e.g., geometric average value, variability) is important for model comparison.

  • Response: In this context we calculate correlations to test the consistencies in between models. You are right, correlations are most meaningful for comparisons of model and data, however it is also suitable for the comparisons in between models. As all models in our study solve the shallow water equations, identical initial conditions should result in almost identical results as long as general model parameters are comparable. Especially in the deep ocean, where even linear equations are valid to a large extent, differences between models are not expected. As a measure to test this consistency we chose correlation coefficients.

    Since validation with observed data is not our goal we did not include RMS Errors with respect to the reference data. These were only included for basic comparisons.

Reference:

Mai, P.M. and Thingbaijam, K.K.S. (2014). SRCMOD: An online database of finite‐fault rupture models. Seismological Research Letters, 85(6), pp.1348-1357. 

Reviewer 3 Report

The observations made in the manuscript are the following:

1. In ec (1) on page 5, indicate what "g" means
2. On page 6, include the table of the experimental design that includes the variables used
3. Indicate in the methodology the program used to carry out the statistical part (correlation).

Author Response

Reply to Reviewer 3

We greatly appreciate the comments from the Reviewer. Please see the following responses to your comments:

Response: 

  1. In ec (1) on page 5, indicate what "g" means
  • Response: This has been added in Line 166.
  1. On page 6, include the table of the experimental design that includes the variables used.
  • Response: We have described the numerical codes main differences. The finite fault solution of the experiment (seismic sources) is mapped and shown in the Supplementary Material. We have extended further explanations on page 7. All sources have been described and the outcomes such as flow depth has been further explained in the Section 2.3. For other variables (e.g. flow depth, bathymetry) that were assessed, please refer to Lines 276-286.
  1. Indicate in the methodology the program used to carry out the statistical part (correlation).
  • Response: This has been included . The data analysis was performed with Matlab (mostly ‘corrcoef’ and ‘polyfit’) and we mentioned the functions that have been used in the text.